# Reaction of Corroles with Sarcosine and Paraformaldehyde: A New Facet of Corrole Chemistry

**DOI:** 10.3390/ijms232113581

**Published:** 2022-11-05

**Authors:** Joana F. B. Barata, Paula S. S. Lacerda, Maria Graça P. M. S. Neves, José A. S. Cavaleiro, Catarina I. V. Ramos, Augusto C. Tomé, Paulo E. Abreu, Alberto A. C. C. Pais

**Affiliations:** 1CESAM, University of Aveiro, 3810-193 Aveiro, Portugal; 2LAQV-REQUIMTE, Department of Chemistry, University of Aveiro, 3810-193 Aveiro, Portugal; 3Department of Chemistry, University of Coimbra, 3004-535 Coimbra, Portugal

**Keywords:** corrole, Mannich-type reaction, metallocorrole, enthalpy, PM7 semi-empirical calculations

## Abstract

Details on the unexpected formation of two new (dimethylamino)methyl corrole isomers from the reaction of 5,10,15-tris(pentafluorophenyl)corrolatogallium(III) with sarcosine and paraformaldehyde are presented. Semi-empirical calculations on possible mechanism pathways seem to indicate that the new compounds are probably formed through a Mannich-type reaction. The extension of the protocol to the free-base 5,10,15-tris(pentafluorophenyl)corrole afforded an unexpected new seven-membered ring corrole derivative, confirming the peculiar behavior of corroles towards known reactions when compared to the well-behaved porphyrin counterparts.

## 1. Introduction

Corroles are tetrapyrrolic macrocycles that have emerged in recent years as an effective and independent outstanding member of the porphyrinoid family [1,2]. These contracted macrocycles, with three hydrogens in the inner core, present remarkable and unique properties, such as the ability to stabilize metal ions in high oxidation states, higher acidity, as well as higher emissions and quantum yields when compared to their porphyrinic counterparts [3,4,5,6]. These features are responsible for corroles’ unusual reactivity and, sometimes, unpredictable behavior, making the corrole macrocycles of great interest in various research areas, namely in the development of tumor-targeting and imaging drugs, antitumoral and antimicrobial photodynamic therapy [7,8,9,10,11,12,13,14,15,16,17,18,19,20,21,22,23,24], sensing [25,26,27,28,29,30], design of light-harvesting systems [31,32,33], and catalysis [4,34,35,36,37,38,39,40,41,42].

The contribution of synthetic chemistry to provide new systems based on corroles is being recognized to have an important impact on the success of those applications. The fine modulation of their electronic and structural features by the preparation of corrole metal complexes [43] and/or by functionalization at the *meso* or β-pyrrolic positions is well-documented in the literature [44,45,46]. In particular, and in sharp contrast with the chemistry of porphyrins [47,48,49], there is much less information concerning the reactivity of corroles in cycloaddition reactions, namely as dipolarophiles in the presence of azomethine ylides [50,51,52]. For instance, treatment of *meso*-tetraarylporphyrins with paraformaldehyde and sarcosine generates, in good yields, pyrrolidine-fused cycloadducts, such as compound **1** (Figure 1) via 1,3-dipolar cycloadditions. Similar cycloaddition reactions also occur with other macrocycles, such as octaphyrin(1.1.1.1.1.1.1.1) [53] and *meso*–*meso*, β-β,β′-β′ triply linked diporphyrins [54]. For the latter ones, instead of a 1,3-dipolar cycloaddition reaction, a regioselective [3+4] cycloaddition occurs, affording cycloadducts of type **2** (Figure 1), with a 2,3,6,7-tetrahydroazepine segment fused at the bay-area across the two porphyrin units.

In 2012, we evaluated the gas-phase behavior of two (dimethylamino)methyl-substituted corrole derivatives [55] obtained from the reaction of 5,10,15-tris(pentafluorophenyl)corrolatogallium(III) with paraformaldehyde and *N*-methylglycine. Here, we report the experimental details of that reaction, the structural characterization of the new compounds, as well as a mechanistic proposal supported by semi-empirical calculations.

In addition, we have extended the same type of reaction to the free-base 5,10,15-tris(pentafluorophenyl)corrole. In this case, we obtained an unexpected new seven-membered ring corrole-type product, confirming the peculiar behavior of corroles towards known reactions.

## 2. Results and Discussion

In order to facilitate the reaction analysis, the results obtained from the reaction of 5,10,15-tris(pentafluorophenyl)corrolatogallium(III) (**3**) and from its free-base 5,10,15-tris(pentafluorophenyl)corrole (**8**) with paraformaldehyde plus sarcosine will be analyzed separately.

### 2.1. Reactions Involving 5,10,15-Tris(Pentafluorophenyl)Corrolatogallium(III)

The first assays involving the reaction of **3** with the ylide generated from paraformaldehyde and sarcosine were performed using the protocol described for *meso*-arylporphyrins [56]. Briefly, to a toluene solution of complex **3**, two equivalents of sarcosine and five equivalents of paraformaldehyde were added. After 5 h at reflux, the thin-layer chromatography (TLC) analysis of the reaction mixture revealed the formation of two new compounds in very small amounts, along with degradation products. After tedious work-up, the two new compounds were isolated and identified by mass spectrometry. The mass spectra of both compounds revealed a molecular ion at *m*/*z* 920 corresponding to the mass of the expected 1,3-dipolar cycloadduct [(M-py)+H]^+^ formed either by the 1,3-dipolar cycloaddition reaction or by the [3+4] cycloaddition reaction. Aiming to favor the formation of these adducts, we repeated the reaction but using only one equivalent of sarcosine and one equivalent of paraformaldehyde. In this way, after 2 h at reflux, the TLC of the reaction mixture showed the presence of the starting corrole and of the latter two new derivatives. Further amounts of 1 equivalent of sarcosine and 1 equivalent of paraformaldehyde were added and the reaction was prolonged for an extra period of 2 h. After 4 h of reaction, the TLC control showed the formation of the expected compounds and a complex mixture of more polar products. To minimize the formation of secondary products, a new reaction was carried out, keeping the temperature at 80 °C. After 4 h of reaction and two extra additions of one equivalent of paraformaldehyde and sarcosine, the TLC control showed, in addition to the starting corrole, the presence of the new derivatives in higher yield and smaller amounts of degradation products. Therefore, after the usual work-up and the separation of the reaction mixture by preparative TLC, it was possible to isolate the two compounds **4** and **5** in 38% and 35% yields, respectively, considering the starting material consumed (Figure 1).

The ^1^H NMR spectrum of the product with a higher Rf value showed, in the aromatic region, four doublets at δ 9.18, 8.73, 8.67, and 8.51 ppm and a singlet at δ 8.64 ppm due to the resonances of seven β-pyrrolic protons and, in the aliphatic region, two singlets at δ 5.04 and 2.61 ppm, which were assigned to the resonances of the CH_2_ and CH_3_ protons, respectively, from the (dimethylamino)methyl group. The coupling constant of the doublets at δ 8.73 and 8.51 ppm, *J* 4.5 Hz, allowed the assignment of these signals to the four β-pyrrolic protons at positions 7, 8, 12, and 13. The doublets at δ 9.18 and 8.67 ppm, with a coupling constant of *J* 4.0 Hz, confirmed that positions 17 and 18 were not substituted. The singlet at δ 8.64 ppm suggested that in position 2 or 3, there was a substituent. However, considering that protons H-2 and H-18 are more deshielded than protons H-3 and H-17 [30,52], the presence of the singlet at δ 8.64 ppm (versus 9.00 ppm of the other derivative) suggested that such resonance was due to the H-3 proton and consequently, that the substitution occurred at position 2. This compound was identified as compound **5**.

The ^1^H NMR spectrum of the product with a lower Rf value showed the resonances of seven β-pyrrolic protons as doublets at δ 9.07, 8.74, 8.66, 8.62, 8.52, and 8.48 ppm and as a singlet at δ 9.00 ppm. The presence of the (dimethylamino)methyl group was also evidenced by the two singlets at δ 4.03 and 2.09 ppm due to the CH_2_ and CH_3_ protons, respectively. This compound was identified as isomer **4**.

The position of the (dimethylamino)methyl group on compounds **4** and **5,** besides 1D NMR analysis was also unambiguously established by 2D NMR spectroscopy (see all spectra in Appendix A). The NOESY spectrum of compound **5** (see Appendix A) was crucial to confirm the position of this group. The spectrum shows space proximity between the CH_2_ group at δ 5.04 ppm with the singlet at δ 8.64 due to H-3 and the doublet at δ 9.18 ppm due to H-18.

The isolation of amines **4** and **5** prompted us, in a first analysis, to propose as plausible formation pathways the thermal ring opening of potential cycloadducts **6** or **7** (Figure 2, pathways (i) and (ii), respectively). However, this type of cleavage is very unlikely. Although there are some examples in the literature for pyrrolidines, this type of ring opening was never observed in clorins nor in other types of porphyrinoids, such as porpholactones, when reacted with ylides [57].

To clarify the possible pathways, theoretical calculations on the enthalpies of formation for compounds **4** and **5** and of the possible intermediates **6** and **7** were performed, and the obtained results are shown in Table 1 and Figure 2. For comparison, the formation enthalpy of the starting corrole **3** is also presented. It should be recalled that locating the transition state is extremely hard due to the intrinsic degrees of freedom. As such, we instead took the simpler route of directly inspecting the stabilities of the compounds, which can be related to the activation barriers, from linear free-energy relationships. This analysis can be subsequently coupled with mechanistic suggestions for defining the pathway.

Inspecting Table 1, it is seen that the relative ordering of compound stability is the same in the presence or absence of the solvent. It is also seen that the enthalpy values for compounds **4** and **5** are similar. However, a significant difference is observed in the formation enthalpies for the potential intermediates: for compound **6**, the formation enthalpy is much lower than for compounds **4** and **5**; thus, if the formation of the (dimethylamino)methyl derivatives was via ring opening of the 1,3-dipolar cycloadduct obtained from **3**, compound **6** could be isolated. On the other hand, compound **7** was shown to be the less stable of the potential intermediate cycloadducts, which suggests that a [3+4] cycloaddition reaction is not favored. Additionally, this pathway could not justify the formation of product **4**. 

Based on these pieces of information, we envisage that the (dimethylamino)methylation of corrole **3** is probably occurring via a Mannich-type reaction, where isomers **4** and **5** result from a nucleophilic attack by the corrole at the iminium ion formed from paraformaldehyde and sarcosine. This is accompanied by a decarboxylation process (Figure 3). Interestingly, this is an example of the potential of azomethine ylides to engage in non-pericyclic reactions, as mentioned by Seidel [58].

### 2.2. Reactions Involving 5,10,15-Tris(Pentafluorophenyl)Corrole

The first assays using the free-base corrole **8** were performed by adding *N*-methylglycine (1 equivalent) and paraformaldehyde (1 equivalent) to 5,10,15-tris(pentafluorophenyl)corrole **8** in dry toluene (0.5 mL). After 2 h at reflux, the TLC of the reaction mixture revealed, in addition to a small amount of the starting corrole, a complex mixture of products. This fact prompted us to repeat the reaction but using 2 equivalents of sarcosine and 5 equivalents of paraformaldehyde, in the same volume of toluene, according to the conditions reported for porphyrins [60]. After 3 h, the TLC of the reaction mixture showed the total consumption of the starting corrole and the presence of a green and very polar product accompanied by several minor ones. After the work-up and chromatographic purification, we were able to isolate the green compound, which was identified by NMR and mass spectrometry (vide infra) as being corrole derivative **9** (Figure 4). Compound **9** was obtained in 19% yield, while the amount of the minor products was not enough for their characterization.

Given the particular structure of compound **9**, we tried to improve the reaction outcome by varying some reaction conditions, such as the volume of the solvent and the sarcosine/paraformaldehyde ratio. Interestingly, we found that the yield of **9** could be greatly improved by performing the reactions at the same ratio of sarcosine and paraformaldehyde, but under more diluted conditions. Just by doubling the volume of toluene from 1.5 to 3 mL, the yield increased to 57% and then to 70% and 73% for volumes of 6 and 9 mL, respectively. Under these conditions, the reactions took 5 h until the starting corrole amount was completely consumed and compound **9** was precipitated directly from the reaction flask with hexane, followed by filtration and subsequent purification.

The identification of compound **9** was based on a careful analysis of its mass spectra and NMR studies (see Appendix A). The mass spectrum showed the molecular ion at *m*/*z* 866, which does not correspond to the mass of the 1,3-dipolar cycloadduct or to the mass of the free-base analogous to amino derivatives **4** or **5** (*m*/*z* 853). Curiously, the ^1^H NMR spectrum (see Appendix A) showed only five signals: one singlet at δ 8.04 ppm and two doublets at δ 8.65 and δ 8.49 ppm in the aromatic zone, corresponding to the resonance of six β-pyrrolic protons, and one broad signal at δ 5.75 ppm and one singlet at δ 1.31 ppm in the aliphatic zone, corresponding to the resonance of ten aliphatic protons.

Considering the NMR and mass spectra, the new derivative suggests the addition of an azomethine ylide plus a methyl group to the corrolic unit. The HMBC spectrum (see Appendix A) showed correlation of the singlet at δ 8.04 ppm with: (a) the signal at δ 69.9 ppm corresponding to the CH_2_ resonances, (b) with δ 119.4 ppm corresponding to C-2 and C-18 resonances, and (c) with δ 134.3 and 140.1 ppm, belonging to quaternary carbon resonances. Thus, and given the long-distance correlation observed between this singlet and carbons C-2 and C-18, the singlet δ 8.04 ppm was attributed to the resonances of the H-3 and H-17 protons and confirmed, through an HSQC spectrum (see Appendix A), that the broad singlet at δ 5.75 ppm refers to the resonance of four aliphatic protons of CH_2_ type. Low-temperature studies were also carried out to distinguish the two protons of the same CH_2_ carbon. In this case, the ^1^H NMR spectrum at 10 °C (see Appendix A) presented two signals at δ 6.08 and 5.50 ppm, whose integrations confirmed the resonance of two CH_2_-type protons. Thus, it was possible to unequivocally indicate the signals corresponding to the protons and carbons of the CH_2_ groups, confirming that the C-2 and C-18 β-pyrrolic positions are substituted.

The possibility that the cycloadduct is predominantly in the zwitterionic structural form **9** can be justified by the high *N*–H acidity of free-base corroles and consequently to a high sensitivity to basic anions [30,61]. In fact, and contrary to what happened with the starting corrole **8**, no changes were detected in the absorption spectrum of the cycloadduct in the presence of fluoride anion, and that might be due to the absence of the third *N*H (see Appendix A). A different situation occurred with the addition of acetic acid, where an inversion in the intensity of the Q bands was only observed in the absorption spectrum of the adduct. The initial spectrum was recovered after the addition of triethylamine (see Appendix A). The theoretical calculations on the formation enthalpies of the zwitterionic structural form of compound **9** were performed by considering the three possible tautomers, with the one presented in Figure 4 being the more stable (see Appendix A); for comparison the formation enthalpy of the starting corrole **8** is presented in Appendix A.

Regarding the formation of compound **9** and considering the previous formation of amines **4** and **5** through Mannich reactions, a possible pathway can involve the quaternization of the tertiary amine with paraformaldehyde, followed by an intramolecular electrophilic aromatic substitution, as indicated in Figure 5.

The presence of the protonated molecular ion at *m*/*z* 866 in the mass spectrum of compound **9** (see Appendix A) supports the proposed structure. Interestingly, this ion justifies various product ions, namely the ion at *m*/*z* 823 that merits special attention due to its high relative abundance. The formation of this ion can be justified by the loss of CH_3_N=CH_2_ (Figure 6). The combined loss of this fragment and a methyl radical gives rise to the formation of the ions with *m*/*z* 808. Other less abundant ions were also observed, probably formed from combined losses of CH_3_N=CH_2_ plus one to six HF molecules, giving rise to the ions at *m*/*z* 803, 783, 763, 743, 723, and 703, respectively (Appendix A). The formation of product ions from successive losses of HF where already reported for other corrole derivatives and for porphyrins bearing pentafluorophenyl groups [62,63,64,65].

## 3. Materials and Methods

### 3.1. Materials

All chemicals and solvents used herein were obtained from commercial sources and were used without further purification, except toluene which was dried using standard procedures and pyrrole which was distilled before used.

All solvents used in mass spectrometry experiments were purchased from commercial sources and used as received.

### 3.2. Techniques

^1^H NMR spectra were recorded at 300.13 or 500.13 MHz and ^13^C NMR spectra at 75.47 or 125.77 MHz. All spectra were recorded at room temperature, but for compound **9**, a ^1^H NMR spectrum was also recorded at 10 °C, as already stated. CDCl_3_ (with drops of C_5_D_5_N) or CD_3_COCD_3_ were used as solvents and TMS as the internal reference. The chemical shifts are expressed in δ (ppm) and the coupling constants (*J*) in hertz (Hz). Unequivocal ^1^H assignments were performed with the aid of two-dimensional NOESY spectra (mixing time of 800 ms), while ^13^C assignments were performed with the two-dimensional (^1^H/^13^C) HSQC and HMBC (delays for long-range *J* C/H couplings were optimized for 7 Hz) experiments. ^13^C spectra of compounds **4** and **5** were improved by internal projection of corresponding HSQC and HMBC experiments.

Preparative thin-layer chromatography was carried out on 20 × 20 cm glass plates coated with silica gel (1 mm-thick).

ESI mass spectra were acquired with a Micromass Q-Tof 2 (Micromass, Manchester, UK), operating in the positive ion mode, equipped with a Z-spray source, an electrospray probe, and a syringe pump. Source and desolvation temperatures were 80 and 150 °C, respectively. Capillary voltage was 3000 V. The spectra were acquired at a nominal resolution of 9000 and at a cone voltage of 30 V. Nebulization and collision gases were N_2_ and Ar, respectively. The flow rate used was 10 µL min^−1^. Solutions of the corroles in methanol (approximately 10^−6^ M) were used.

Product-ion spectra were acquired by selecting the precursor ions with the quadrupole and using the hexapole as a collision cell, with energies from 20 to 80 eV.

Accurate mass measurements were performed at a resolution of 9600 FWHM (full-width at half maximum) using the protonated molecule of 5-pentafluorophenyl-10,15,20-triphenylporphyrin (monoisotopic mass 900.1245) as a reference ion.

### 3.3. Theoretical Calculations for Model Structures

The calculations were carried out on Intel-based computers, running Linux, using MOPAC2016 [59]. The geometry of the various structures was optimized with MOPAC2016. All the calculations in MOPAC2016 used the PM7 [66,67,68] Hamiltonian, with COSMO (Conductor-like Screening Model) [68] for the implicit solvent model using toluene. For this simulation, we used 2.38 as the dielectric constant for toluene and the effective radius of the solvent molecule used was 1.3 Å, as suggested by Klamt [69]. Each structure was drawn from the corresponding 2D structure, and then minimized. All geometry optimizations were performed such that the gradient was less than 0.01 kcal/Å. Afterwards, the nature of the critical point was determined from the observation of the eigenvalues of the Hessian matrix (matrix containing the second derivative of the energy in relation to the nuclear coordinates): for a minimum, all eigenvalues should be real, and that was the case for each obtained geometry. On all images, atom colors are: hydrogen (white), carbon (gray), nitrogen (blue), fluorine (green), and gallium (magenta).

### 3.4. Experimental Procedure

The synthetic details for the preparation of 5,10,15-tris(pentafluorophenyl)corrole **8** and 5,10,15-tris(pentafluorophenyl)corrolatogallium(III)(pyridine) 3 are provided in the literature [70,71].

#### 3.4.1. General Procedure for the Synthesis of Compounds **4** and **5**

A toluene (0.5 mL) solution of 5,10,15-tris(pentafluorophenyl)corrolatogallium(III)(pyridine) (21.3 mg, 22.6 μmol), sarcosine (1.1 equiv., 2.3 mg, 25.8 μmol), and paraformaldehyde (1.0 equiv., 0.8 mg, 22.6 μmol) was heated at 80 °C for 2 h under a nitrogen atmosphere. Additional portions of 1 equivalent sarcosine and 1 equivalent paraformaldehyde were added, and the reaction mixture was heated for another 2 h. After being cooled to room temperature, the separation and purification of the new compounds were carried out on preparative TLC using ethyl acetate/petroleum ether/pyridine (50:65:1) as an eluent to afford, by decreasing order of R_f_, the starting corrole **3** (13.6 mg), compound **5** (2.9 mg, 13% (35% based on the consumed starting material)), and compound **4** (3.1 mg, 14% (38% based on the consumed starting material)).

3-(Dimethylamino)methyl-5,10,15-tris(pentafluorophenyl)corrolategallium(III)(pyridine) (**4**): ^1^H NMR (300 MHz, CDCl_3_, and few drops of C_5_D_5_N): δ 9.07 (d, 1H, *J* 4.0 Hz, H-18), 9.00 (s, 1H, H-2), 8.74 (d, 1H, *J* 4.5 Hz, H-β), 8.66 (d, 1H, *J* 4.0 Hz, H-17), 8.62 (d, 1H, *J* 4.6 Hz, H-β), 8.52 (d, 1H, *J* 4.5 Hz, H-β), 8.48 (d, 1H, *J* 4.6 Hz, H-β), 4.03 (s, 2H, CH_2_), 2.09 (s, 6H, 2xCH_3_). ^13^C NMR (based on HSQC and HMBC projections): δ 142.6, 140.7, 139.4, 134.7, 133.7, 127.1 (2xC-β), 124.7 (C-17), 123.3 (2xC-β), 120.2 (C-2), 118.0 (C-18), 58.5 (CH_2_), 45.3 (2xCH_3_). HRMS (ESI) calculated for C_40_H_16_N_5_F_15_Ga [(M-py)+H]^+^ 920.0422, found 920.0413.

2-(Dimethylamino)methyl-5,10,15-tris(pentafluorophenyl)corrolategallium(III)(pyridine) (**5**): ^1^H NMR (300 MHz, CDCl_3_, and a few drops of C_5_D_5_N): δ 9.18 (d, 1H, *J* 3.9 Hz, H-18), 8.73 (d, 2H, *J* 4.5 Hz, H-β), 8.67 (d, 1H, *J* 3.9 Hz, H-17), 8.64 (s, 1H, H-3), 8.51 (d, 2H, *J* 4.5 Hz, H-β), 5.04 (s, 2H, CH_2_), 2.61 (s, 6H, 2xCH_3_). ^13^C NMR (based on HSQC and HMBC projections): δ 149.4, 143.2, 140.6, 127.4, 125.0 (C-5,6,9,10,11,14,15), 135.2 (C-1,19), 128.9 (C-4), 127.4 (2xC-β), 125.0 (C-17, C-3), 123.4 (2xC-β), 118.4 (C-18), 56.7 (CH_2_), 45.0 (2xCH_3_). HRMS (ESI) calculated for C_40_H_16_N_5_F_15_Ga [(M-py)+H]^+^ 920.0422, found 920.0425.

#### 3.4.2. General Experimental Procedure for Compound **9**

To a solution of sarcosine (2 equiv.) and paraformaldehyde (5 equiv.) in dry toluene (9 mL), 5,10,15-tris(pentafluorophenyl)corrole (20 mg, 25.1 μmol) was added. The reaction mixture was heated at 80 °C for 5 h under a nitrogen atmosphere. After being cooled to room temperature, hexane was added, and the resulting precipitate was filtered. The solid was then purified by preparative TLC using ethyl acetate/chloroform (9:1) as an eluent. Compound **9** was obtained in 73% yield (15.5 mg).

Compound **9**: ^1^H NMR (300.13 MHz, CD_3_COCD_3_): δ 8.65 (d, 2H, H-7,13, or H-8,12, *J* 4.6 Hz), 8.49 (d, 2H, H-7,13, or H-8,12, *J* 4.6 Hz), 8.04 (s, 2H, H-3,17), 5.75 (broad signal, 4H, CH_2_), 1.31 (s, 6H, CH_3_). ^1^H NMR (500.13 MHz, at 10 °C, CD_3_COCD_3_): δ 8.65 (d, 2H, H-7,13, or H-8,12, *J* 4.4 Hz), 8.49 (d, 2H, H-7,13, or H-8,12, *J* 4.4 Hz), 8.05 (s, 2H, H-3,17), 6.08 (broad signal, 2H, CH_2_), 5.50 (broad signal, 2H, CH_2_), 1.29 (s, 6H, CH_3_). ^13^C NMR (CD_3_COCD_3_, 125.77 MHz): 143.3 (C-9,11 or C-6,14), 142.8 (C-9,11 or C-6,14), 140.1 (C-4,16 or C-1,19), 134.3 (C-4,16 or C-1,19), 126.6 (C-7,13 or C-8,12), 125.6 (C-7,13 or C-8,12), 120.1 (C-3,17), 119.4 (C-2,18), 69.9 (2xCH_2_), 29.0 (2xCH_3_). UV-Vis CH_3_COCH_3_ nm (log ε): 426 (4.83) 594 (3.98) 623 (4.42). MS (ESI) *m*/*z* 866 [M+H]^+^. HRMS (ESI) calculated for C_41_H_19_N_5_F_15_ [M+H]^+^ 866.13955, found 866.13631.

## 4. Conclusions

Contrary to what occurs for porphyrins, 5,10,15-tris(pentafluorophenyl)corrolatogallium(III) reacts with paraformaldehyde and sarcosine to afford, as major products, two isomeric corroles, with one (dimethylamino)methyl group at the 2- or 3-position of the corrole core. We propose that the new corrole isomers are formed through a Mannich-type reaction, never described before in corrole chemistry. Surprisingly, under similar experimental conditions, the free-base 5,10,15-tris(pentafluorophenyl)corrole affords the unprecedented seven-membered ring derivative **9**.

## Data Availability

Not applicable.

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
