# Peer review of "Reaction of Corroles with Sarcosine and Paraformaldehyde: A New Facet of Corrole Chemistry"

_ijms, 2022, doi:10.3390/ijms232113581_

Round 1

Reviewer 1 Report

This is an intersting report on unusual formation of azepine-fused corrole derivative. In my view, the finding of the authors is worth publication, but several issues should be resolved before that.

(i) The most important issue is the proposed structure of compound 9, which is not quite fit to the reported NMR data. The authors attribute the signals at 134.2 and 140.8 ppm in 13C NMR to C-1 and C-19, which have almost identical chemical surrounding. The only difference between them is the location of the N-H proton at the opposite side of the corrole core, 5 bonds or ca. 0.5 nm away from them. Is it enough to secure the observed 6 ppm difference between the signals? In compound 5, according to the data in section 4 (p. 10, line 295), these carbons have identical chemical shifts, though their chemical surrounding differs much stronger. Besides, only two doublets appear in the aromatic range of 1H NMR spectrum of compound 9. This suggests averaging of the position of the NH protons due to prototropic equilibria, but this also means averaging of the field for carbons C-1 and C-19. So the question here is what is the reason for such strong difference of chemical shift for C-1 and C-19?

(ii) The authors report compound 9 as a cation. What is the counterion? Can it be a zwitter-ionic compound (e.g., N-deprotonated) rather than a cation?

(iii) The authors report a dynamic phenomenon in 1H NMR of compound 9. This is quite interesting. The CH2-protons become much differentiated at -10 C (ca. 0.68 ppm difference in chemical shifts in 1H NMR). What is the source of this asymmetry?

(iv) The authors suggest that the formation of compounds 4 and 5 cannot proceed through intermediate 6 as this intermediate has lower enthalpy of formation which would allow its isolation. This is not necessarily the case, as it is the activation barrier on going from 6 to 4 (E6-4) or from 6 to 5 (E6-5) and the activation barrier on going from 3 to 6 (E3-6 )that matter. If E3-6>>E6-4(5) then intermediate 6 could not be isolated. Yet, the proposed by the authors mechanism of formation of 4 and 5 through Mannich reaction looks reasonable.

(iv) Experimental part, line 319: compound 9 is a salt, why the molecular ion in MS is described as a radical-cation?

(v) Please provide separate descriptions of 1H NMR spectra of compound 9 at 25 C and at -10 C. The current presentation is quite confusing and incomplete. Please add the low temperature spectra to the supplementary information. Please calibrate the 2D spectra in the supplementary information.

(vi) Section 3.3, line 272: should it be "positive" instead of "real"? Section 3.2, line 240: is "CD3COCD3" missing here?

(vii) References 10-15 are not mentioned in the text.

Author Response

This is an interesting report on unusual formation of azepine-fused corrole derivative. In my view, the finding of the authors is worth publication, but several issues should be resolved before that.

We thank the reviewer for recognizing the value of our work and for his or her valuable comments.

The most important issue is the proposed structure of compound 9, which is not quite fit to the reported NMR data. The authors attribute the signals at 134.2 and 140.8 ppm in 13C NMR to C-1 and C-19, which have almost identical chemical surrounding. The only difference between them is the location of the N-H proton at the opposite side of the corrole core, 5 bonds or ca. 0.5 nm away from them. Is it enough to secure the observed 6 ppm difference between the signals?

We thank the reviewer for this alert and the observation is absolutely correct since there is no evidence for a 6 ppm difference in the chemical shift of C-1 and C-19. This is a mistake in the NMR description of compound 9 based on HMBC spectra. In fact, the HMBC spectra showed correlation of the singlet at δ 8.04 ppm with: a) the signal at δ 69.9 ppm corresponding to the CH2 resonances; b), with δ 119.4 ppm corresponding to C-2,18 resonances and c) with δ 134.3 e 140.1 ppm belonging to quaternary carbon resonances. For the sake of clarity and since there is no evidence for other assignments, we corrected the 13C NMR spectrum and its description as:

13C NMR (CD3COCD3, 125.77 MHz): 143.3 (C-9,11 or C-6,14), 142.8 (C-9,11 or C-6,14), 140.1 (C-4,16 or C-1,19), 134.3 (C-4,16 or C-1,19), 126.6 (C-7,13 or C-8,12), 125.6 (C-7,13 or C-8,12), 120.1 (C-3,17), 119.4 (C-2,18), 69.9 (2xCH2), 29.0 (2xCH3).

In compound 5, according to the data in section 4 (p. 10, line 295), these carbons have identical chemical shifts, though their chemical surrounding differs much stronger. Besides, only two doublets appear in the aromatic range of 1H NMR spectrum of compound 9. This suggests averaging of the position of the NH protons due to prototropic equilibria, but this also means averaging of the field for carbons C-1 and C-19. So the question here is what is the reason for such strong difference of chemical shift for C-1 and C-19?

As we stated before we agree with the reviewer that there is no reason for such difference. In the new version the assignment of the quaternary carbons was corrected.

(ii) The authors report compound 9 as a cation. What is the counterion? Can it be a zwitter-ionic compound (e.g., N-deprotonated) rather than a cation?

We thank the reviewer for this pertinent comment and suggestion. Considering the proposed mechanism, we believe that the counterion is the deprotonated sarcosine (CH3NHCH2COO-). However, having in account the NH acidity of corroles (please see, Chemical Physics Letters [61]) it is possible that an equilibrium with the zwitterionic derivative can occur. In order to confirm this, we decided to analyze how the visible absorption spectrum of the adduct was affected in acidic conditions and in the presence of TBAF [30].

The following text was added to the manuscript:

“The possibility that the cycloadduct is predominantly in the zwitterionic structural form 9 can be justified by the high N–H acidity of free-base corroles and consequently to an high sensitivity to basic anions [30,61]. In fact, and contrary to what happened with the starting corrole 8, no changes were detected in the absorption spectrum of the cycloadduct in the presence of fluoride anion (F-) and that might be due to the absence of the third NH (see Figure S20-A, Supporting Information). A different situation occurred with the addition of acetic acid, where an inversion in the intensity of the Q bands was only observed in the absorption spectrum of the adduct; the initial spectrum was recovered after the addition of triethylamine (see Figure S20-B, Supporting Information). The theoretical calculations on the formation enthalpies of the zwitterionic structural form of compound 9 were performed by considering the three possible tautomers, being the one presented in scheme 4 the more stable (see Table S2, Supporting Information).”

So, in the new version we take into account that equilibrium and the proposal that compound 9 is principally in the zwitterionic form.

(iii) The authors report a dynamic phenomenon in 1H NMR of compound 9. This is quite interesting. The CH2-protons become much differentiated at -10 C (ca. 0.68 ppm difference in chemical shifts in 1H NMR). What is the source of this asymmetry?

We thank the reviewer for allowing us to clarify this consideration. We just like to point out that the spectrum was obtained at 10 ºC and not at -10 ºC, as written.

In our point of view, the different chemical environment of the CH2 protons might related to some distortions on corrole planarity. In this way, the NMR at 10 °C facilitated the resolution of the CH2 peaks that were overlapped at room temperature NMR.

(iv) The authors suggest that the formation of compounds 4 and 5 cannot proceed through intermediate 6 as this intermediate has lower enthalpy of formation which would allow its isolation. This is not necessarily the case, as it is the activation barrier on going from 6 to 4 (E6-4) or from 6 to 5 (E6-5) and the activation barrier on going from 3 to 6 (E3-6) that matter. If E3-6>>E6-4(5) then intermediate 6 could not be isolated. Yet, the proposed by the authors mechanism of formation of 4 and 5 through Mannich reaction looks reasonable.

We thank the reviewer for allowing us to clarify this consideration since we agree with the comment. In fact, in our proposal the ring-opening of the pyrrolidine ring would occur in a two steps-pathway starting from 3 which affords firstly the potential intermediate 6 and then the amino derivatives 4 and 5. In order to avoid misunderstandings we considered in table 1, for comparison, the formation enthalpy of the starting corrole 3 which confirms that 6 is more stable than the starting corrole 3. Also, in order to be more clear, we add the following information” if the formation of the (dimethylamino)methyl derivatives 4 and 5 were formed via ring opening of the 1,3-dipolar cycloadduct 6, then compound 6 could be isolated."

(iv) Experimental part, line 319: compound 9 is a salt, why the molecular ion in MS is described as a radical-cation?

Considering that compound 9 is predominantly in the zwitterionic form the m/z value of 866 corresponds to the protonated molecular ion [M+H]+.

(v) Please provide separate descriptions of 1H NMR spectra of compound 9 at 25 C and at -10 C. The current presentation is quite confusing and incomplete. Please add the low temperature spectra to the supplementary information. Please calibrate the 2D spectra in the supplementary information.

We thank the reviewer for this comment and suggestion. The following descriptions of 1H NMR spectra of compound 9 at 25 ºC and at 10 ºC were added. The 1H NMR spectrum at 10 ºC is in the supplementary information.

Compound 9: 1H NMR (300.13 MHz, CD3COCD3): δ 8.65 (d, 2H, H-7,13 or H-8,12, J 4.6 Hz), 8.49 (d, 2H, H-7,13 or H-8,12, J 4.6 Hz), 8.04 (s, 2H, H-3,17), 5.75 (broad signal, 4H, CH2), 1.31 (s, 6H, CH3). 1H NMR (500.13 MHz, at 10 °C, CD3COCD3): δ 8.65 (d, 2H, H-7,13 or H-8,12, J 4.4 Hz), 8.49 (d, 2H, H-7,13 or H-8,12, J 4.4 Hz), 8.05 (s, 2H, H-3,17), 6.08 (broad signal, 2H, CH2), 5.50 (broad signal, 2H, CH2), 1.29 (s, 6H, CH3).

(vi) Section 3.3, line 272: should it be "positive" instead of "real"? Section 3.2, line 240: is "CD3COCD3" missing here?

We thank the reviewer for the detailed revision. These corrections were made in the new version of the manuscript.

(vii) References 10-15 are not mentioned in the text.

These references were added to the main text.

Reviewer 2 Report

1. It is a solid manuscript reporting the synthesis and thorough characterization of new corrole derivatives. Data acquired by semi-empirical calculations allowed to describe possible reaction paths and to suggest a Mannich-type reaction. Compound 9 was synthesized with a very high yield by dilution experiments.

Only a few formal corrections are needed.

1. Scheme 3: indicate No 3 of corrole!

2. English usage (corrected)

i) Mannich-type reaction (thoughout)

ii)

- line 16: Mannich-type reaction – see also additional examples!

- line 17: corrole with a seven-membered ring (this sounds better)

- line 71: described for meso-arylporphyrins

- line 153: Based on these pieces of information…

- line 188: careful analysis; 190: free bases

- line 233: All chemicals; 236: All solvents

- line 327: unprecedented

This is not a full list – seek the help of a professional!

Author Response

Reviewer 2

  1. It is a solid manuscript reporting the synthesis and thorough characterization of new corrole derivatives. Data acquired by semi-empirical calculations allowed to describe possible reaction paths and to suggest a Mannich-type reaction. Compound 9 was synthesized with a very high yield by dilution experiments.

Only a few formal corrections are needed.

  1. Scheme 3: indicate No 3 of corrole!
  2. English usage (corrected)
  3. i) Mannich-type reaction (thoughout)

ii)

- line 16: Mannich-type reaction – see also additional examples!

- line 17: corrole with a seven-membered ring (this sounds better)

- line 71: described for meso-arylporphyrins

- line 153: Based on these pieces of information…

- line 188: careful analysis; 190: free bases

- line 233: All chemicals; 236: All solvents

- line 327: unprecedented

This is not a full list – seek the help of a professional!

We thank the reviewer for recognizing the value of the work. All the above comments were corrected according to the suggestions. Also, other corrections were considered after a careful reading from an English native speaker.

Reviewer 3 Report

The paper discusses the reaction of 5,10,15-tris(pentafluorophenyl)corrole and its gallium complex with paraformaldehyde and sarcosine. The result of the reactions is dramatically changed for these two corroles giving in the first case azepane and Mannich product in the second case. Although the authors carry out quantum mechanical calculations of these reactions, they do not give an answer to the cause of the formation of different reaction products. The structure of the three compounds obtained has been poorly proven, it is necessary to add at least the data of mass spectra. In addition, the number of signals in the 1H and 13C NMR spectra does not coincide with the calculated data; it is necessary to prove more clearly the difference in the structure of isomers 4 and 5. The paper contains a few typos and other shortcomings.

1.      There are no references 10-15 in the text of the paper.

2.      The total yield of products in scheme 1 significantly exceeds 100%.

3.      There is no anion in the ionic structure of compound 9.

4.      The experimental part does not specify the yield of the initial corolle when obtaining compounds 4 and 5, as well as the yield of compound 9 in mgs.

The paper can be recommended for publication in the journal ‘International Journal of Molecular Sciencesafter major revision.

Author Response

Reviewer 3

The paper discusses the reaction of 5,10,15-tris(pentafluorophenyl)corrole and its gallium complex with paraformaldehyde and sarcosine. The result of the reactions is dramatically changed for these two corroles giving in the first case azepane and Mannich product in the second case. Although the authors carry out quantum mechanical calculations of these reactions, they do not give an answer to the cause of the formation of different reaction products.

We agree with the reviewer that depending on the corrole used (free-base or the gallium complex) the main compounds isolated and characterized were different. However, in both cases, the formation of amines 4 and 5 and compound 9 were justified via a Mannich reaction.

Regarding the formation of compound 9 an extra quaternization of the tertiary amine with paraformaldehyde followed by an intramolecular electrophilic aromatic substitution occurs. Probably the more flexibility of the free-base macrocycle can facilitate this intramolecular reaction which is favoured with higher reaction dilution. It is important to refer that other products are obtained but unfortunately in amounts that did not allow us to characterize them.

The structure of the three compounds obtained has been poorly proven, it is necessary to add at least the data of mass spectra.

The data related to the high-resolution mass spectrum of compound 9 was added to the supporting information.

Regarding compounds 4 and 5, reference [55] reports an extensive mass spectrometry studies devoted to the differentiation of these two isomers. The mass spectra data of these two compounds has been already published (Ramos, C.I. V.; Graça Santana-Marques, M.; Ferrer-Correia, A.J.; Barata, J.F.B.; Tomé, A.C.; Neves, M.G.P.M.S.; Cavaleiro, J.A.S.; Abreu, P.E.; Pereira, M.M.; Pais, A.A.C.C. Differentiation of aminomethyl corrole isomers by mass spectrometry. J. Mass Spectrom. 2012, 47, 516–522, doi:10.1002/jms.2973).

In addition, the number of signals in the 1H and 13C NMR spectra does not coincide with the calculated data; it is necessary to prove more clearly the difference in the structure of isomers 4 and 5. The paper contains a few typos and other shortcomings.

In the revised version we clarify better the difference in the structures of isomers 4 and 5 and correct some missing data related with such matter.

We also corrected typos and shortcomings that were meanwhile suggested or found by us. About the number of signals in 1H and 13C NMR spectra some of them comprise more than one proton or carbon due to their similitude.

  1. There are no references 10-15 in the text of the paper.

We thank the reviewer for this alert and the references [10-15] were added to the manuscript

  1. The total yield of products in scheme 1 significantly exceeds 100%.

We thank the reviewer for this alert.

The sentence was rephrased, in order to avoid misunderstandings. In fact, the yields were calculated considering the corrole consumed in the reaction and not the recovered starting corrole.

“So, after the usual work-up and the separation of the reaction mixture by preparative TLC, it was possible to isolate the two compounds 4 and 5 in 38% and 35% yields, respectively, considering the starting material consumed (Scheme 1).”

  1. There is no anion in the ionic structure of compound 9.

Considering the proposed mechanism, we believe that the counterion is the deprotonated sarcosine (CH3NHCH2COO-). However, having in account the NH acidity of corroles (please see, Chemical Physics Letters [61]) it is possible that an equilibrium with the zwitterionic derivative can occur. In order to confirm this possibility, we decided to analyze how the visible absorption spectrum of the adduct was affected in acidic conditions and in the presence of TBAF [30].

The following text was added to the manuscript:

“The possibility that the cycloadduct is predominantly in the zwitterionic structural form 9 can be justified by the high N–H acidity of free-base corroles and consequently to a high sensitivity to basic anions [30,61]. In fact, and contrary to what happened with the starting corrole 8, no changes were detected in the absorption spectrum of the cycloadduct in the presence of fluoride anion (F-) and that might be due to the absence of the third NH (see Figure S20-A, Supporting Information). A different situation occurred with the addition of acetic acid, where an inversion in the intensity of the Q bands was only observed in the absorption spectrum of the adduct; the initial spectrum was recovered after the addition of triethylamine (see Figure S20-B, Supporting Information). The theoretical calculations on the formation enthalpies of the zwitterionic structural form of compound 9 were performed by considering the three possible tautomers, being the one presented in scheme 4 the more stable (see Table S2, Supporting Information).”

So, in the new version we take into account that equilibrium and the proposal that compound 9 is principally in the zwitterionic form is put forward.

4.The experimental part does not specify the yield of the initial corrole when obtaining compounds 4 and 5, as well as the yield of compound 9 in mgs.

According with the suggestion, the amount of the starting corrole when obtaining compounds 4 and 5 and the amount of compound 9 were added to the experimental part in the new version of the manuscript

Round 2

Reviewer 1 Report

The authors have addressed my concerns, and I am glad to recommend this work for publication in IJMS.

Some typos found:

- line 218: "134.3 e 140.1 ppm" -> "134.3-140.1 ppm"

- line 228: "ß-pyrrolic" -> "b-pyrrolic"

- line 319: "real" ->"positive"

Reviewer 3 Report

The authors took into account all the comments of the reviewer, and the article can be published in this form after careful editing.